# L-Type Amino Acid Transporter 1 (LAT1) Promotes PMA-Induced Cell Migration through mTORC2 Activation at the Lysosome

**DOI:** 10.3390/cells12202504

**Published:** 2023-10-23

**Authors:** Kun Tae, Sun-Jick Kim, Sang-Woo Cho, Hoyeon Lee, Hyo-Sun Cha, Cheol-Yong Choi

**Affiliations:** Department of Biological Sciences, Sungkyunkwan University, Suwon 16419, Republic of Korea; xorjssla01@gmail.com (K.T.); godmous79@hanmail.net (S.-J.K.); swcho0628@naver.com (S.-W.C.); hoyeonzigy@gmail.com (H.L.); hyosnin@naver.com (H.-S.C.)

**Keywords:** LAT1, mTORC2, PMA, cell migration

## Abstract

The mTOR signaling pathway integrates signaling inputs from nutrients, including glucose and amino acids, which are precisely regulated by transporters depending on nutrient levels. The L-type amino acid transporter 1 (LAT1) affects the activity of mTORC1 through upstream regulators that sense intracellular amino acid levels. While mTORC1 activation by LAT1 has been thoroughly investigated in cultured cells, the effects of LAT1 expression on the activity of mTORC2 has scarcely been studied. Here, we provide evidence that LAT1 recruits and activates mTORC2 on the lysosome for PMA-induced cell migration. LAT1 is translocated to the lysosomes in cells treated with PMA in a dose- and time-dependent manner. Lysosomal LAT1 interacted with mTORC2 through a direct interaction with Rictor, leading to the lysosomal localization of mTORC2. Furthermore, the depletion of LAT1 reduced PMA-induced cell migration in a wound-healing assay. Consistent with these results, the LAT1 N3KR mutant, which is defective in PMA-induced endocytosis and lysosomal localization, did not induce mTORC2 recruitment to the lysosome, with the activation of mTORC2 determined via Akt phosphorylation or the LAT1-mediated promotion of cell migration. Taken together, lysosomal LAT1 recruits and activates the mTORC2 complex and downstream Akt for PMA-mediated cell migration. These results provide insights into the development of therapeutic drugs targeting the LAT1 amino acid transporter to block metastasis, as well as disease progression in various types of cancer.

## 1. Introduction

The L-type amino acid transporter 1 (LAT1) imports large neutral amino acids, such as leucine and phenylalanine, or aromatic amino acids in exchange for intracellular glutamine or other non-essential amino acids. This exchange occurs in a pH- and Na^+^-independent manner [1,2,3]. The LAT1-mediated influx of amino acids is crucial for various cellular activities, including cell growth, proliferation, and metabolic maintenance. In particular, leucine is an effective amino acid that stimulates protein synthesis via the mTOR signaling pathway. LAT1 forms a heterodimer with 4F2hc (4F2 antigen heavy chain; CD98 heavy chain), and this heterodimerization is essential for amino acid transporter function, structural stability, and the plasma membrane localization of LAT1 [4]. The protein expression levels of LAT1 have frequently been observed to be elevated in various cancer cells, including breast [5,6,7,8,9], liver [10,11,12,13,14], lung [15,16,17,18], and prostate [19,20,21]. Increased protein levels of LAT1 can serve as a biomarker for malignant cancer, and several reports have discussed its prognostic correlation in terms of predicting surgical outcomes [22,23,24].

The mammalian target of the rapamycin (mTOR) signaling pathway integrates both intracellular and extracellular signals, serving as a central regulator of cell metabolism, growth, proliferation, and survival [25]. The mTOR pathway is activated by various factors, including growth factors, cellular energy, and nutrients such as amino acids. The Rag and Rheb GTPases play distinct yet essential roles in activating the mTORC1 pathway. The Rags control the subcellular localization of mTORC1, while Rheb stimulates its kinase activity. Moreover, Rheb integrates signaling inputs from growth factors and glucose, while the Rag GTPase senses signaling inputs related to amino acid levels [26]. Higher amino acid levels activate the Rag GTPases, leading to the recruitment of mTORC1 to the lysosomal surface, where it becomes concentrated. Rheb is also localized to the lysosomal surface. During growth factor withdrawal, the TSC tumor suppressor inhibits mTORC1 by promoting GTP hydrolysis by Rheb. As a result, both Rag and Rheb inputs ensure mTORC1 activation only when both are active [27]. The localization of mTORC1 to the lysosomes is critical for its ability to sense and respond to fluctuations in amino acid levels. Lysosomes play a significant role beyond merely facilitating the proper assembly of the mTORC1 regulatory pathway, and the activities of mTORC1 and lysosomes are closely intertwined [28]. The levels of amino acids within the lysosomal lumen are sensed by the amino acid transporter SLC38A9, which directly modulates mTORC1 activity through the vacuolar H^+^-adenosine triphosphatase (v-ATPase) [29,30,31]. LAT1 is also localized to the lysosome and participates in mTORC1 activation [32]. However, it remains unclear whether LAT1-mediated amino acid efflux from the lysosome is necessary for mTORC1 activation in a manner similar to SLC38A9.

mTORC2 is another mTOR complex composed of subunits shared with mTORC1 (Deptor and mLST8) and distinct subunits, such as Rictor (rapamycin-insensitive companion of mTOR) and mSin1. Rictor is known as the scaffold protein for substrate binding to mTORC2. mTORC2 can phosphorylate AKT at Ser473 to cause its maximum activation [33,34,35,36]. The activation of mTORC2 occurs in response to growth factors, nutrients, or hormones [33,37], and it mediates the activation of several targets, including AKT, SGK1, and PKC [33,36,38]. These targets function in protein synthesis, cell survival, actin organization, and cell migration. mTORC2 regulates the actin cytoskeleton through PKC and plays various regulatory roles in cell migration through cytoskeletal remodeling [39]. While the impact of LAT1 function on mTORC1 activation has been extensively studied, the functional relationship between LAT1 and mTORC2 activation, especially at the lysosomal surface, has not yet been investigated.

In this study, we present evidence demonstrating that upon the treatment of cells with phorbol 12-myristate 13-acetate (PMA), LAT1 is translocated to the lysosome and recruits mTORC2 to facilitate Akt activation, subsequently inducing cell migration. Notably, the N-terminal ubiquitination of LAT1 is implicated in PMA-triggered endocytosis leading to lysosomal localization, where LAT1 interacts with Rictor to activate mTORC2 and facilitate Akt phosphorylation. Furthermore, our findings reveal that PMA-induced cell migration is orchestrated via the endocytosis of LAT1, wherein lysosomal LAT1 recruits the mTORC2 complex to activate Akt. 

## 2. Materials and Methods

### 2.1. Cell Culture and Transfection

HEK293T (human embryonic kidney) and HeLa (human cervix adenocarcinoma) cells were purchased from the American Type Culture Collection (ATCC). HEK293T and HeLa cells were cultured in Dulbecco’s modified Eagle’s medium (DMEM) supplemented with 10% fetal bovine serum (Gibco, Waltham, MA, USA) and 1% penicillin-streptomycin solution. The cells were maintained at 37 °C in a humidified atmosphere with 5% CO_2_. HeLa cells stably expressing HA-tagged LAT1 and HA-tagged TMEM192 were generated via puromycin selection for 72 h. Cells were transfected with plasmid DNA using Lipofectamine 3000, while siRNA was transfected using Lipofectamine RNAiMAX, following the manufacturer’s protocols.

### 2.2. Plasmids

Human LAT1 cDNA was purchased from the Korea Human Gene Bank (Clone ID: hMU005138). LAT1 was amplified through PCR and subsequently inserted into the *BamHI* and *EcoRV* sites of either the pCS5 + 3XHA or pCS5 + 3X Flag vector, kindly provided by Dr. Jaewhan Song, Yonsei University, Korea. To generate ubiquitination-defective mutants of LAT1, namely LAT1 N3KR (K19R, K25R, K30R) and LAT1 C4KR (479R, K481R, K483R, K498R), the Muta-DirectTM Site-Directed Mutagenesis Kit (iNtRON, #15071) was employed. The pLJC5-Tmem192-3xHA plasmid, created by David Sabatini, was obtained from Addgene (#102930).

### 2.3. Antibodies

Anti-S6K (#9202), anti-phospho S6K Thr389 (#9234), anti-AKT (#9272), anti-phospho AKT Ser473 (#9271), anti-LAT1 (#5347), anti-mTOR (#2983), anti-Raptor (#2280), anti-Flag (#14793), and anti-Rictor (#2140) were purchased from Cell Signaling Technology. Anti-tubulin (05-829) was purchased from Millipore. Anti-β-actin (A700-057) and anti-Flag M2 (F1804) were purchased from Sigma. HA-horseradish peroxidase (HRP) (11-814-150-001) was purchased from Roche. Anti-HA (sc-7392) was purchased from Santa Cruz Biotechnology. Anti-LAMP2 (ab25631) and anti-sodium potassium ATPase (ab76020) were purchased from Abcam.

### 2.4. Immunoblotting

Cells were lysed using ice-cold lysis buffer containing 1% NP-40, 137 mM of NaCl, 20 mM of Tris-HCl pH 8.0, 2 mM of EDTA, 10 mM of NaF, 1 mM of Na_3_VO_4_, and a protease inhibitor cocktail from Sigma. The cell lysates were incubated on ice for 20 min and subsequently separated into soluble fractions through centrifugation at 13,000× rpm for 10 min. The resulting supernatant was combined with 5 × Laemmli sample buffer and heated to 100 °C for 10 min. Samples were resolved using SDS-PAGE and transferred onto a PVDF membrane (EMD Millipore, Burlington, MA, USA). The membranes were blocked with 4% skim milk for 30 min, followed by overnight incubation with primary antibodies. Afterward, the membranes were washed three times with TBST and probed with appropriate secondary antibodies for 1 h. Subsequently, the membranes were washed three times with TBST and visualized using an ECL substrate (iNtRON, #16028).

### 2.5. Immunoprecipitation

Cells were lysed utilizing an ice-cold lysis buffer consisting of 0.3% CHAPS, 40 mM of Tris-HCl pH 7.4, 120 mM of NaCl, 2 mM of EDTA, 40 mM of HEPES pH 7.4, 10 mM of NaF, 1 mM of Na_3_VO_4_, and a protease inhibitor cocktail. The cell lysates were incubated on ice for 30 min, and the soluble fraction was separated through centrifugation at 13,000× rpm for 10 min. The resulting supernatant was combined with the specified primary antibody and incubated at 4 °C for 12 h on a rotator. Subsequently, the mixture was further incubated for an additional 2 h with protein G agarose beads (GenDEPOT, Katy, TX, USA). The beads were subjected to 5 washes using the lysis buffer and then eluted with 2.5 × Laemmli sample buffer. The immunoprecipitated samples were subjected to analysis through immunoblotting using the designated antibodies.

### 2.6. Immunofluorescence and Proximity Ligation Assay (PLA)

HeLa cells were cultured on coverslips. Following PMA treatment at the specified time and dose, the cells were fixed using 3% paraformaldehyde for 10 min. Subsequently, the cells were permeabilized using 0.1% Triton X-100 in PBS for 5 min, followed by a 30-min blocking step utilizing 2% BSA in PBS. The fixed cells were subjected to a 2-h incubation with the designated primary antibody, after which they were further incubated with an appropriate fluorescence-conjugated secondary antibody for 20 min at room temperature. To mount the cells, mounting media from Sigma (DUO82040) was used. In the case of Duolink^®^ PLA Fluorescence, the cells underwent the same fixation and permeabilization procedure as immunofluorescence, adhering to the manufacturer’s instructions from Sigma (DUO92101). Fluorescence microscopy was conducted using a Zeiss LSM 700 confocal microscope along with ZEN 2012 software (version 1.1.2.0). The acquired images were subsequently processed using Adobe Photoshop.

### 2.7. Quantitation of Plasma Membrane Proteins Using Biotinylation

Cells cultured in 60 mm plates were employed for the biotinylation of each sample. The cells were subjected to three rinses using ice-cold PBS and subsequently incubated with PBS containing 1 mg/mL of sulfo-NHS-biotin (APExBIO, Houston, TX, USA) for 20 min. The reaction was quenched by treating the cells with PBS containing 150 mM glycine, followed by an additional three rinses. The cells were then harvested into a 1.5 mL tube and lysed using the same lysis buffer employed for immunoblotting. After lysis, samples were subjected to centrifugation at 13,000× rpm for 10 min. The resulting supernatant was transferred to a new 1.5 mL tube and incubated with streptavidin magnetic beads (Thermo Scientific, Waltham, MA, USA, 88816) for 8 h. The immunoprecipitates were washed five times with lysis buffer and subsequently eluted using 2.5 × Laemmli sample buffer.

### 2.8. Lysosome Immunoprecipitation

The immunopurification of lysosomes (Lyso-IP) was conducted following previously described methods with slight adjustments [40]. Cells cultured in 100 mm plates were employed for each Lyso-IP. The cells underwent three PBS rinses and were then collected in a 1.5 mL tube using PBS containing a protease inhibitor cocktail. Afterward, the cells were homogenized through 30 strokes of a homogenizer and subsequently centrifuged at 3000× rpm for 2 min. The supernatant, containing cellular organelles including lysosomes, was transferred to a new 1.5 mL tube and subjected to incubation with anti-HA magnetic beads (Thermo Scientific, 88836) for 15 min at 4 °C. The immunoprecipitates were washed three times with PBS and subsequently treated with 1% Triton-X 100. The samples were separated from the beads utilizing a magnetic stand. The eluted lysosomal proteins were combined with 2.5 × Laemmli sample buffer and then utilized for immunoblotting using the designated antibodies.

### 2.9. Wound-Healing Assay

Cells were cultured to confluence in a 6-well culture plate. A single line scratch wound was generated using the SPLScarTM Scratcher (SPL, Cat. 201906). The wounded monolayers were washed once with PBS to eliminate detached cells and debris and subsequently incubated in medium containing either DMSO or 1 µM of PMA. Cell migration was assessed at each designated time point post-scratching and quantified using Image J (v1.54g) software. Each experiment was repeated three times.

### 2.10. Transwell Migration Assay

A549 cells were transfected with the indicated siRNA before seeding with Transwell plates (6.5 mm Transwell with a 8.0 µm pore polycarbonate membrane insert; Corning Costar, Kennebunk, ME, USA). Cells (1 × 10^5^) in 200 µL of 1% FBS culture media were seeded in the upper chamber, and 600 µL of 10% FBS culture media in the presence or absence of PMA was placed in the lower chamber. Following incubation for 16 h, cells were fixed with 3% paraformaldehyde and permeabilized with 0.1% Triton-X 100. Cells were stained with hematoxylin, and cells adhering to the upper chamber were removed using a cotton swab. Stained cells were counted under a light microscope. 

### 2.11. Statistical Analysis

Statistical significance was analyzed via one-way ANOVA, followed by Bonferroni’s multiple comparison test using GraphPad Prism 5 software. *p* values < 0.05 were considered statistically significant.

## 3. Results

### 3.1. PMA Induces LAT1 Endocytosis to the Lysosome in a Time- and Dose-Dependent Manner

LAT1 is a bidirectional amino acid transporter responsible for the transport of neutral amino acids and the regulation of mTORC1 activity by facilitating the entry of leucine into the cytoplasm [41]. Despite its localization to both the plasma membrane and lysosomes, the specific lysosomal function of LAT1 remains unclear. In order to investigate the lysosomal function of LAT1, we utilized HeLa cell lines expressing HA-tagged LAT1 and employed phorbol 12-myristate 13-acetate (PMA), a compound that stimulates PKC activity, to promote the internalization of LAT1 into the lysosomes [42]. HeLa cells expressing HA-LAT1 were exposed to 1 µM of PMA, and the subcellular localization of LAT1 was assessed using anti-HA antibody staining at various time points to identify the optimal conditions for studying the lysosomal function of LAT1. Under normal culture conditions, HA-LAT1 predominantly localized to the plasma membrane. However, the administration of PMA led to a gradual translocation of HA-LAT1 to the lysosomes over time (Figure 1A). Colocalization analysis revealed a 2.5-fold increase in LAT1-Lamp2 colocalization after 6 h of PMA treatment (Figure 1B), concomitant with a reduction in plasma membrane-localized LAT1 (Figure 1A). This experiment was also conducted using varying concentrations of PMA. Increasing concentrations of PMA, up to 1 µM, elicited a concentration-dependent shift of LAT1 from the plasma membrane to the lysosome (Figure 1C,D). To validate the movement of LAT1 to the lysosome following PMA treatment, lysosomes were isolated from cell lines expressing HA-Tmem192 [40], which serves as a lysosome marker. Western blot analysis of the isolated lysosomes demonstrated an increase in lysosome-associated endogenous LAT1 levels 6 h after PMA treatment (Figure 1E). Taken together, these findings indicate that PMA administration induces the endocytosis of LAT1 in a time- and dose-dependent manner, leading to its relocation to the lysosomes.

### 3.2. LAT1 N-Terminal Ubiquitination Is Crucial for PMA-Induced LAT1 Endocytosis

Nedd4L-mediated LAT1 ubiquitination has been implicated in LAT1 endocytosis [42]. LAT1 is composed of twelve transmembrane domains, and seven regions, including the N- and C-terminal domains, are exposed to the cytoplasm. The N-terminal domain contains three lysine residues (18K, 25K, and 30K), while the C-terminal domain harbors four lysine residues (479K, 481K, 483K, and 498K). To investigate the role of ubiquitination, LAT1 N3KR and C4KR mutants were generated by substituting the three N-terminal lysine residues and four C-terminal lysine residues with arginine, respectively. The impacts of these mutations on PMA-induced LAT1 endocytosis and lysosomal targeting were assessed. While wild-type LAT1 and the C4KR mutant exhibited movement from the plasma membrane to the lysosome upon PMA treatment, the N3KR mutant remained localized to the plasma membrane (Figure 2A,B). To further verify the N3KR defect in PMA-induced endocytosis, lysosomes were isolated from cells expressing wild-type LAT1, N3KR, or C4KR mutants in the presence or absence of PMA treatment. Consistent with the immunostaining results, lysosome-associated LAT1 increased with wild-type LAT1 and the C4KR mutant upon PMA treatment, but it did not increase with the N3KR mutant (Figure 2C). Additionally, the levels of plasma membrane-localized LAT1 were assessed using the biotinylated plasma membrane (PM) fraction. Western blot analysis of the PM fraction demonstrated a reduction in the PM levels of wild-type LAT1 and the C4KR mutant following PMA treatment, whereas this effect was not observed with the N3KR mutant (Figure 2D). Collectively, these findings underscore the essential role of N-terminal ubiquitination in facilitating LAT1 endocytosis to the lysosome in response to PMA treatment. 

### 3.3. LAT1 Mediates mTOR Recruitment to the Lysosomes

Given that the mechanism of LAT1 involvement in mTOR activation is yet to be elucidated, and considering the known activation of mTOR at the lysosomal surface [28], we sought to explore whether LAT1 mediates mTOR activation at the lysosome. To investigate this possibility, we assessed mTOR recruitment to isolated lysosomes following varying durations of PMA treatment. Western blot analysis of isolated lysosomes revealed simultaneous increases in the recruitment of both LAT1 and mTOR to the lysosome in a time-dependent manner upon PMA treatment (Figure 3A). To further validate the in vivo association between LAT1 and mTOR in response to PMA, we conducted a proximity ligation assay (PLA). This assay generates fluorescence signals from amplified closed circular DNA when LAT1 and mTOR are found in close proximity. PLA experiments employing anti-mTOR and anti-HA antibodies demonstrated a significant enhancement in fluorescence signals in conjunction with wild-type LAT1 following PMA treatment. A similar observation was made with the C4KR mutant (Figure 3B,C), suggesting that both wild-type LAT1 and the C4KR mutant translocate to the lysosome, where mTOR complexes are recruited to the lysosomal surface. In contrast, the endocytosis-defective N3KR mutant did not display an increase in fluorescence signals (Figure 3B,C), emphasizing the necessity of the lysosomal localization of LAT1 via endocytosis for mTOR recruitment to the lysosomes.

### 3.4. LAT1 Regulates mTORC2 Signaling on the Lysosome

mTOR exists in two conserved and structurally distinct kinase complexes referred to as mTORC1 and mTORC2. Each complex consists of both common and distinct subunits, allowing them to phosphorylate distinct sets of substrates, thereby regulating various cellular processes, including cell growth, proliferation, and cytoskeletal organization for migration [43]. Raptor is a representative distinct subunit of mTORC1, while Rictor fulfills this role in mTORC2. To investigate the potential association of lysosomal LAT1 with a specific mTOR complex, immunoprecipitation was conducted using anti-Raptor and anti-Rictor antibodies in the presence or absence of PMA treatment. Western blot analysis revealed that LAT1 was present in immunoprecipitates with the anti-Rictor antibody, but it was not present with the anti-Raptor antibody, indicating that LAT1 recruits the mTORC2 complex to the lysosomal surface (Figure 4A). Moreover, the interaction between LAT1 and Rictor increased upon PMA treatment, suggesting that LAT1 interacts with the mTORC2 complex through Rictor at the lysosomal surface. To elucidate the functional link between the LAT1-mTORC2 complex and mTORC2 activation, Akt phosphorylation, a hallmark of mTORC2 activity, was assessed. The treatment of cells with PMA induced Akt phosphorylation, which was attenuated upon the depletion of LAT1 (Figure 4B, lanes 2 and 4), implying a requirement of LAT1 for mTORC2-mediated Akt phosphorylation. Intriguingly, the reintroduction of wild-type LAT1 into LAT1-depleted cells reinstated Akt phosphorylation following PMA treatment (Figure 4B, lane 6), highlighting the role of LAT1 in inducing mTORC2-mediated Akt phosphorylation. Consistent with the PLA results (Figure 3B), the introduction of the C4KR mutant into LAT1-depleted cells restored Akt phosphorylation (Figure 4B, lane 10), while this effect was not observed with the N3KR mutant, which lacks lysosomal targeting (Figure 4B, lane 8). These findings collectively prove that LAT1 recruits mTORC2 to the lysosomal surface, leading to the activation of downstream targets such as Akt.

### 3.5. LAT1-Mediated mTORC2 Activation Affects Cell Migration

Akt activation plays a crucial role in wound healing and cell migration [44,45,46,47]. To assess the functional significance of the interplay between LAT1 and mTORC2, a wound-healing assay was conducted to monitor cell migration. A single linear scratch was generated using mock- or LAT1-depleted HeLa cells, and wound healing was monitored in the presence or absence of PMA. An analysis of the wounded area revealed a notable enhancement in the wound closure of HeLa cells in the presence of PMA. In contrast, wound healing of LAT1-depleted HeLa cells was considerably delayed (Figure 5A,B), underscoring the requirement of LAT1 for PMA-induced wound healing. In addition, the transwell assay confirmed that PMA-induced cell migration was inhibited via LAT1 or Rictor depletion (Figure 5C,D). Given that the LAT1 N3KR mutant exhibited defective Akt phosphorylation (Figure 4B), wound-healing assays were performed using HeLa cells expressing either the LAT1 N3KR or C4KR mutant. Endogenous LAT1 was depleted using siRNA targeting the LAT1 3′-UTR in HeLa cells expressing HA-LAT1 WT, N3KR, or C4KR to eliminate the function of endogenous LAT1. Subsequently, wound-healing assays were carried out in the presence or absence of PMA. Microscopic examination revealed efficient wound closure in cells expressing wild-type LAT1 and the C4KR mutant, whereas wound healing in N3KR-expressing cells was significantly delayed (Figure 5E,F). This observation suggests that LAT1 promotes cell migration by orchestrating the lysosomal activation of mTORC2 and subsequent downstream signaling, including Akt activation.

## 4. Discussion

The mTOR signaling pathway is active in most human cancers, where it drives cell growth, proliferation, and cancer metabolism. The activities of mTOR are regulated by diverse signaling inputs, including glucose, growth factors, and cellular amino acid levels, which are modulated by numerous amino acid transporters [48]. Each amino acid transporter imports or exports a specific set of amino acids with varying affinities. LAT1, for instance, transports large neutral amino acids, such as leucine, isoleucine, phenylalanine, methionine, tyrosine, histidine, tryptophan, and valine [49]. Among these examples, leucine holds particular significance as it plays a pivotal role in activating mTORC1 by recruiting it to the lysosomal surface. Leucine exerts its effect via interactions with various regulatory proteins, including components of the GATOR complexes. Consequently, LAT1 has been identified as being overexpressed in various types of cancer, and research into LAT1 inhibitors as potential drug targets for cancer treatment is underway. High expression of LAT1 has been associated with poor prognosis in non-small cell lung cancer, pancreatic cancer, brain tumors, prostate cancer, and breast cancer. These observations underscore the relationship between LAT1 expression and cancer malignancy [7,15,21,22,50,51,52]. Furthermore, LAT1 has been suggested to play a significant role in the metastatic process of various human cancers [5]. An analysis of LAT1 expression was conducted between the primary site and a concordant pulmonary metastatic site in 93 cancer patients, including those with colon cancers, breast cancers, head and neck cancers, genital cancers, and soft-tissue sarcomas. This analysis revealed that LAT1 expression was notably higher in the metastatic sites compared to the primary sites. Correspondingly, in renal cell carcinoma (RCC), patients with high LAT1 expression levels exhibited shorter overall survival (OS) and progression-free survival (PFS) compared to those with low LAT1 expression levels [53]. In our present study, we have demonstrated that LAT1 interacts with Rictor, a component of the mTORC2 complex, leading to Akt phosphorylation through mTORC2 activation (Figure 4). Co-immunoprecipitation experiments conducted with anti-Raptor and anti-Rictor antibodies showed that Raptor did not exhibit interaction with LAT1, whereas Rictor interacted with LAT1 in a manner dependent on lysosomal LAT1. These findings suggest that LAT1 triggers distinct mechanisms for activating mTORC1 and mTORC2. Specifically, mTORC2 is directly activated through a physical interaction with LAT1 via Rictor, whereas mTORC1 may be activated indirectly through mTORC1 regulators such as GATOR1/2, contingent on intracellular amino acid levels. The physical interaction between LAT1 and Rictor, along with the activation of downstream targets of mTORC2, provides a foundational molecular understanding of the pivotal role played by LAT1 in cancer cell migration (Figure 5), the elevated LAT1 expression in metastatic sites compared to primary sites [5], and the lower survival rates observed in patients with higher LAT1 expression levels [53]. 

The lysosomal localization of the mTORC1 complex has undergone extensive investigation, leading to the identification of its functional upstream regulators, which include Ragulator, Rag GTPases, and Rheb GTPases [54,55]. The recruitment of the mTORC1 complex to the lysosomal surface represents a critical step for mTORC1 activation, influencing the sensing of amino acid levels and regulation of autophagy. Several studies have indicated that mTORC2 is also recruited to the lysosomal surface. The immunoblotting of isolated lysosomes has revealed the presence of mTORC2 complex components, including mSin1, Rictor, and Akt, on the lysosomal surface [56]. Lysosomal mTORC2 is implicated in the regulation of chaperone-mediated autophagy (CMA). Under basal conditions, mTORC2 signaling inhibits the translocation complex through the Akt-mediated phosphorylation of GFAP, thereby controlling the baseline level of CMA activity in most cells [57]. LAT1 is localized not only to the plasma membrane but also to the lysosomes. While LAPTM4b has been identified as a lysosomal membrane protein responsible for recruiting LAT1 to the lysosomes, previous studies have primarily focused on LAT1 function in V-ATPase-dependent mTORC1 activation [32]. The role of lysosomal LAT1 is only beginning to be comprehended. In our current study, we have demonstrated that lysosomal LAT1 mediates the activation of mTORC2 and its downstream target Akt, ultimately leading to the induction of cell migration. To explore this observation, we employed the LAT1 N3KR mutant, which is a lysosomal targeting-defective mutant of LAT1 due to point mutations in N-terminal ubiquitination sites. The LAT1 N3KR mutant exhibited defects in PMA-induced endocytosis and subsequent trafficking to the lysosome, as well as lysosomal association with mTORC2. These findings indicate that LAT1 localizes to the lysosome, where it activates the mTORC2 complex. The plasma membrane was long thought to serve as the primary site for mTORC2 activation. Recent research, however, employing a reporter system to monitor endogenous mTORC2 activity in specific subcellular compartments, revealed distinct mTORC2 populations at the mitochondria, early and late endosomes, and plasma membrane [58]. While it is conceivable that mTORC2 is activated on the lysosomal surface in response to autophagy regulation and the levels of lysosomal luminal amino acids, further investigation is necessary to elucidate the precise mechanisms underlying mTORC2 activation at the lysosome.

mTORC2 has been known to induce cell migration by directly phosphorylating and activating several actin cytoskeleton regulators, including Rho GTPases and PKC-α [59]. However, in this study, PKC activation via the treatment of cells with PMA induced mTORC2 activation at the lysosome via the lysosomal LAT1 amino acid transporter. Therefore, PKC can also induce mTORC2 activation, leading to the further activation of PKC-mediated actin cytoskeleton reorganization for cell migration. Moreover, both mTORC1 and mTORC2 can phosphorylate and stabilize the Nedd4L E3 ubiquitin ligases [60], which are responsible for the ubiquitination of the LAT1 N-terminus. Consequently, mTORC2 activation may enhance the lysosomal localization of LAT1, further contributing to mTORC2 activation. The potential positive feedback loops between mTORC2 and PKC- or Nedd4L-mediated LAT1 ubiquitination serve to reinforce the mTORC2-driven induction of cell migration. While this study utilized the synthetic chemical PMA to demonstrate the LAT1-mediated mTORC2 activation and subsequent induction of cell migration, these phenomena could potentially be reproduced under the physiological conditions of PKC activation in the tumor microenvironment. Unraveling the underlying mechanisms that modulate the balance of LAT1 levels between the plasma membrane and the lysosome would offer crucial insights into the role of LAT1 in tumor progression and metastasis. 

## 5. Conclusions

We demonstrate, for the first time, that the L-type amino acid transporter LAT1 recruits mTORC2 to the lysosomal surface for PMA-induced cell migration through a direct interaction with Rictor, an mTORC2-specific subunit. While the LAT1-mediated importing of large neutral amino acids, especially leucine, plays a crucial role in the recruitment of mTORC1 to the lysosome and mTORC1-mediated cell growth and proliferation, LAT1-mediated recruitment of mTORC2 to the lysosome, which leads to cell migration, provides another layer of complexity in the coordination between mTORC1 and mTORC2. Further studies regarding the effects of the amino acid transporter activity of lysosomal LAT1 on mTORC1 and mTORC2 are required to elucidate the function of lysosomal LAT1 and for the development of drugs targeting LAT1. 

## Figures and Tables

**Figure 1 cells-12-02504-f001:**
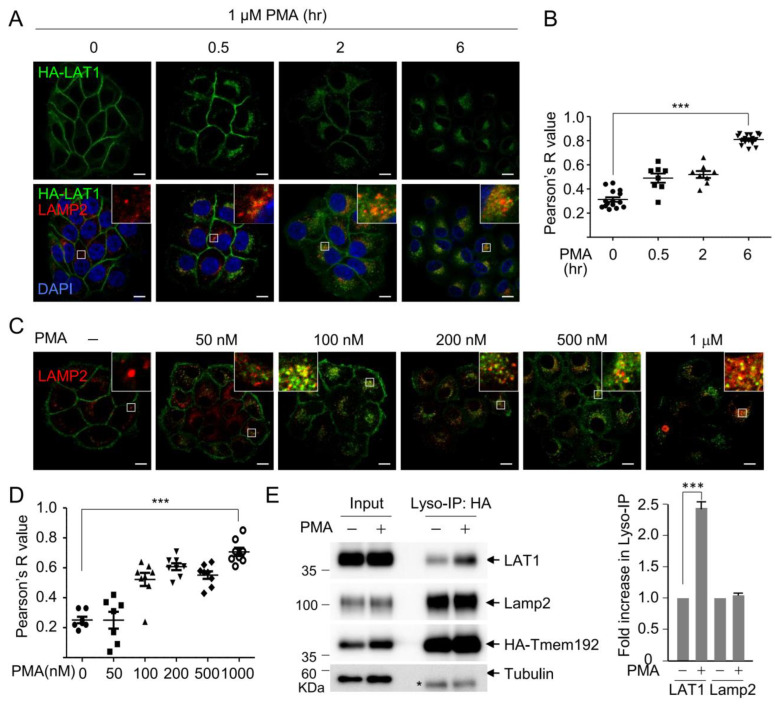
PMA induces LAT1 endocytosis in a time- and dose-dependent manner. (**A**) A HeLa HA-LAT1 cell line was treated with 1 µM of PMA at the indicated times. Each cell was subjected to immunostaining using anti-HA and anti-LAMP2 antibodies, with nuclei stained using DAPI. The inset is a zoom-in image. Scale bar: 10 µm. (**B**) Colocalization between HA-LAT1 and LAMP2 was determined through Pearson’s correlations (graphs represent mean ± standard deviation). Statistical analysis was performed utilizing one-way ANOVA, followed by Bonferroni’s multiple comparison test, where *** *p* < 0.001. (**C**) HeLa HA-LAT1 cell line treated with PMA for 6 h exhibited endocytosis of HA-LAT1 in a concentration-dependent manner. Each cell was immunostained with anti-HA antibody and anti-LAMP2 antibody, and nuclei were stained with DAPI. Scale bar: 10 µm. (**D**) Colocalization between HA-LAT1 and LAMP2 was determined using Pearson’s correlations (graphs represent mean ± standard deviation). Statistical analysis was performed using one-way ANOVA, followed by Bonferroni’s multiple comparison test, where *** *p* < 0.001. (**E**) HEK293T HA-Tmem192-overexpressing cell line was treated with MOCK or PMA for 6 h, and lysosomes were isolated using an anti-HA antibody. Endogenous lysosomal LAT1 was detected via immunoblotting using anti-LAT1 antibody. The graph depicts densitometric analysis results of Western blots, showing the fold increase in LAT1 during Lyso-IP following PMA treatment. Data are presented as the mean ± SD (*n* = 3). Asterisk indicates statistically significant differences (*** *p*< 0.001). The asterisk indicates a non-specific band.

**Figure 2 cells-12-02504-f002:**
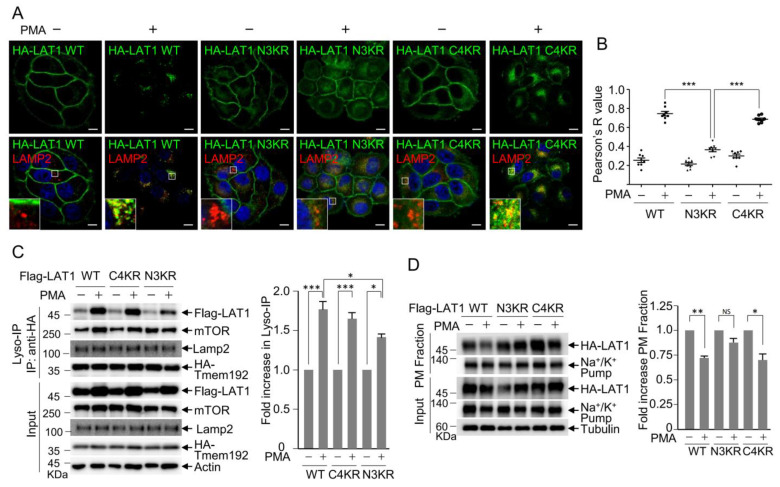
N-terminal ubiquitination is essential for PMA-induced LAT1 localization to the lysosome. (**A**) HeLa HA-LAT1 WT, N3KR, and C4KR expression cell lines were treated with 1 µM of PMA for 6 h. Cells were immunostained with anti-HA antibody and anti-LAMP2 antibody, and nuclei were stained with DAPI. The inset is a zoom-in image. Scale bar: 10 µm. (**B**) Colocalization between HA-LAT1 WT, N3KR, or C4KR and LAMP2 was determined using the Pearson’s correlation test (graphs represent the mean ± standard deviation). Statistical analysis was performed using one-way ANOVA, followed by Bonferroni’s multiple comparison test, where *** *p* < 0.001. (**C**) The HEK293T HA-Tmem192-overexpressing cell lines were transfected with Flag-LAT1 WT-, N3KR-, and C4KR-encoding plasmids. After treating the cells with 200 nM of PMA for 6 h, lysosomal fractions were isolated using Lyso-IP and subjected to immunoblotting with the indicated antibodies. The graph depicts densitometric analysis results of Western blots, showing the fold increase in wild-type LAT1 and LAT1 mutants during Lyso-IP following PMA treatment. Data are presented as the mean ± SD (*n* = 3). Asterisk indicates statistically significant differences (* *p* < 0.05, *** *p* < 0.001). (**D**) HeLa HA-LAT1 WT, N3KR, or C4KR cell lines were treated with 1 µM of PMA for 6 h, and plasma membrane proteins were labeled with biotin. The plasma membrane fraction was isolated through streptavidin pulldown, and the precipitates were subsequently subjected to immunoblotting with the indicated antibodies. The graph depicts densitometric analysis results of Western blots, showing the fold increase in wild-type LAT1 and LAT1 mutants during plasma membrane fractionation following PMA treatment. Data are presented as the mean ± SD (*n* = 3). Asterisk indicates statistically significant differences (* *p* < 0.05, ** *p* < 0.01, NS: not significant).

**Figure 3 cells-12-02504-f003:**
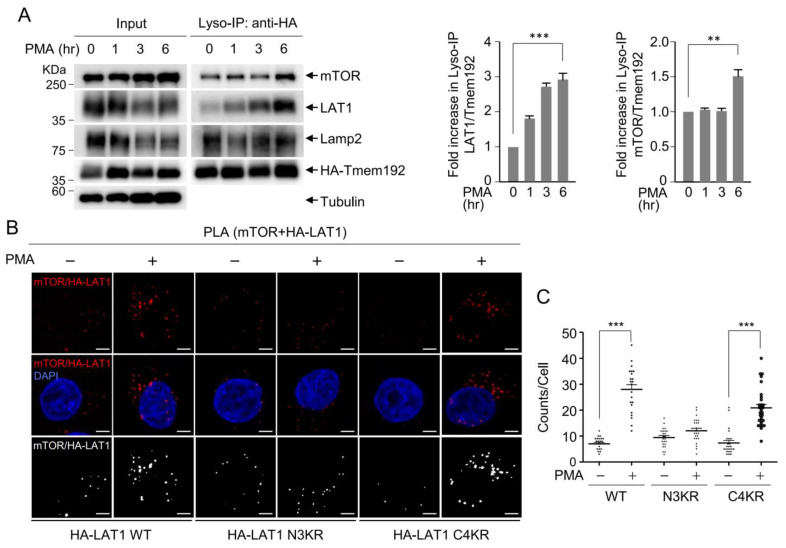
LAT1 N-terminal ubiquitination is crucial for PMA-induced mTOR recruitment to the lysosomes. (**A**) HEK293T HA-Tmem192-overexpressing cell line was treated with 200 nM of PMA for the indicated times. Lysosomal fractions were isolated using Lyso-IP. The immunoprecipitates were subjected to immunoblotting using the indicated antibodies. The graph depicts the densitometric analysis results of Western blots, showing the fold increase in LAT1 and mTOR during Lyso-IP following PMA treatment. Data are presented as the mean ± SD (*n* = 3). Asterisk indicates statistically significant differences (** *p* < 0.01, *** *p* < 0.001). (**B**) HeLa HA-LAT1 WT, N3KR, or C4KR cell lines were treated with 1 µM of PMA for 6 h. Each sample was fixed using 3% paraformaldehyde, and a PLA assay was performed with anti-mTOR antibody and anti-HA antibody. The third line presents images in which all pixels whose values lie below the threshold are converted to black, and all pixels with values above the threshold are converted to white. Scale bar: 5 µm. (**C**) Statistical analysis was performed using one-way ANOVA, followed by Bonferroni’s multiple comparison test, where *** *p* < 0.001.

**Figure 4 cells-12-02504-f004:**
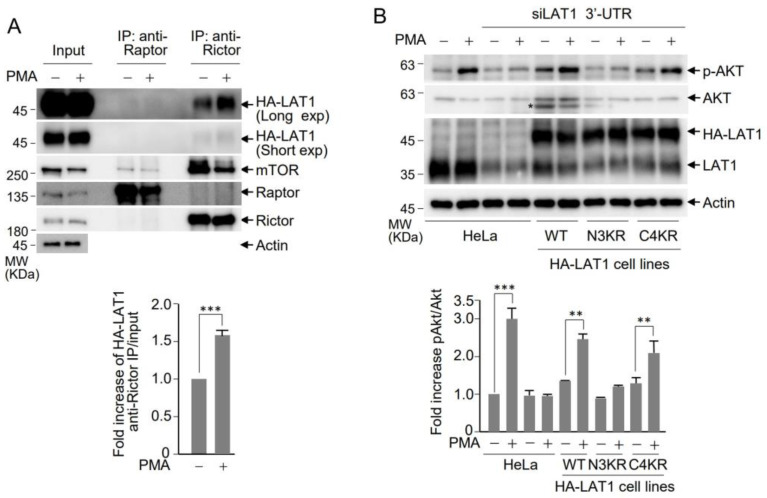
PMA enhances mTORC2 activation through LAT1 interaction with Rictor. (**A**) The HeLa HA-LAT1-expressing cell line was treated with 1 µM of PMA for 6 h, and cell lysates were prepared using 0.3% CHAPS lysis buffer. Cell lysates were immunoprecipitated using anti-Raptor or anti-Rictor antibodies. The immunoprecipitates were subjected to immunoblotting using the specified antibodies. The graph depicts densitometric analysis results of Western blots, showing the fold increase in HA-LAT1 during immunoprecipitation using anti-Rictor antibody. Data are presented as the mean ± SD (*n* = 3). Asterisk indicates statistically significant differences (*** *p* < 0.001). (**B**) HeLa cells, as well as HeLa HA-LAT1 WT-, N3KR-, or C4KR-expressing cell lines, were subjected to the depletion of endogenous LAT1 via the transfection of siRNA targeting the 3′UTR of LAT1 or control siRNA prior to treatment with 1 µM of PMA for 6 h. Cell lysates were analyzed through immunoblotting using the indicated antibodies. The asterisk in the Western blot indicates a non-specific band. The graph depicts densitometric analysis results of Western blots, showing the fold increase in pAkt following PMA treatment. Data are presented as the mean ± SD (*n* = 3). Asterisk indicates statistically significant differences (** *p* < 0.01, *** *p* < 0.001).

**Figure 5 cells-12-02504-f005:**
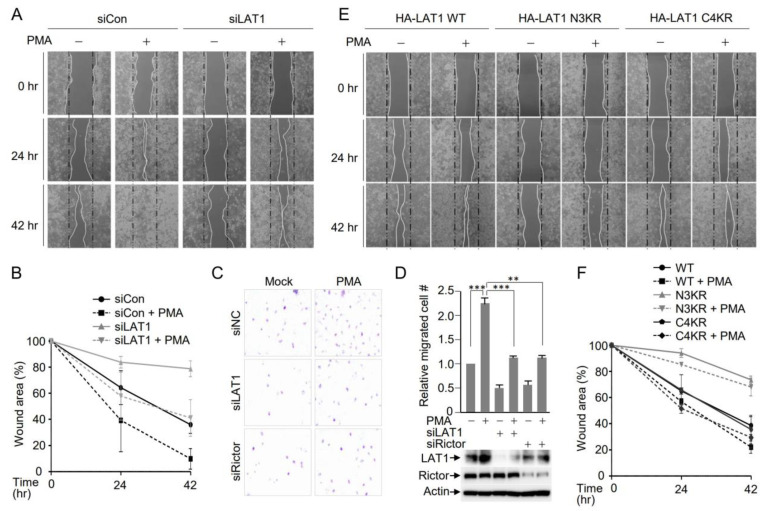
N-terminal ubiquitination of LAT1 is crucial for PMA-induced cell migration. (**A**) HeLa cells were subjected to endogenous LAT1 depletion through siRNA treatment. Subsequently, cells were treated with 1 µM of PMA before initiating a single-line scratch. Illustrated are representative images depicting the progress in wound healing at the indicated time points for unstimulated and 1 µM of PMA-stimulated cells. (**B**) The graph illustrates the quantification of the wounded area measured using the ImageJ program. *n* = 3 independent experiments. (**C**) A549 cells were transfected with siLAT1 or siRictor and subjected to a transwell assay in the presence or absence of PMA. (**D**) The graph indicates the quantitation of migrating cells. Statistical analysis was performed using one-way ANOVA, followed by Bonferroni’s multiple comparison test, where *** *p* < 0.001 and ** *p* < 0.01. *n* = 3 independent experiments. Western blot indicates the knockdown efficiency of LAT1 and Rictor. (**E**) HeLa HA-LAT1 WT, N3KR, or C4KR cell lines were depleted of endogenous LAT1 via the transfection of siRNA targeting the 3′UTR of LAT1. Subsequent to depletion, cells were treated with 1 µM of PMA before introducing a single-line scratch. Displayed are representative images showing the progression of wound healing at the indicated time points for unstimulated and 1 µM of PMA-stimulated cells. (**F**) The graph presents the quantified wounded area measured utilizing the ImageJ program. *n* = 3 independent experiments.

## Data Availability

Not applicable. The conclusions outlined in the manuscript are based on relevant datasets available in the manuscript.

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
