# Peer review of "L-Type Amino Acid Transporter 1 (LAT1) Promotes PMA-Induced Cell Migration through mTORC2 Activation at the Lysosome"

_cells, 2023, doi:10.3390/cells12202504_

Round 1

Reviewer 1 Report

This is an interesting manuscript describing LAT1 to recruit mTORC2 to lysosomes and promote PMA-induced migration. A few points should be addressed:

Figure 1E: WB analysis should be quantified. How often were these experiments performed? Are WB results statistically significant? Minor: Please clarify the asterics next to the bottom band in the IP lanes?

Figure 2C-D: Western blots should be quantified. Why do amounts of Flag-LAT1 increase with PMA treatment in Figure 1C, while HA-Lat1 remains rather constant with and without PMA? The amount of Flag-tagged and HA-tagged LAT1 in the IP and Pulldown should be normalized to input. What is the amount of endogenous LAT1 in these experiments? How often were these experiments performed? Are WB results statistically significant?

Figure 3A: Western blots should be quantified. The amount of endogenous LAT1 appears downregulated with PMA (see input 3 and 6 hrs). mTOR levels in the IP using anti-HA do not seem to increase after 3h PMA incubation. If this is the case, the text should be revised accordingly (lane 265-266). How often were these experiments performed? Are WB results statistically significant?

Figure 4A-B: In this experiment it appears mTOR levels are reduced with PMA treatment, while Rictor levels remain unchanged. If Rictor and mTOR are associated in equimolar amounts, does this indicate a pool of Rictor proteins not associated with mTOR? WB analysis should be quantified. How often were these experiments performed? Are WB results statistically significant? Fig. 4B. Total amounts of Akt should be provided and WB quantification of p-Akt should be normalized to total Akt.

Figure 5: Cell migration of HeLa cells after LAT1 knockdown or ectopically expressing LAT1 WT and mutants with and without PMA is shown. Based on the previous experiments, the authors conclude that LAT1 promotes migration via mTORC2. Unless further control experiments (e.g. Rictor knockdown) are provided, the involvement of mTORC1 cannot be excluded and this should be clarified in the Abstract (lane 22-23) and Results (lane 341-343).

Author Response

Response:

- As pointed out by the referee, we have added densitometric quantitation results of Figure 1E and Figure 2C. However, we are very sorry for not being able to provide the quantitation results for other Western blot results. The Western blot analyses presented in this paper were conducted two or three times, and we reproducibly observed the same results. Unfortunately, however, we did not carry out the same set of experiment lane by lane, which is required for statistical analysis of the results. Therefore, we do not have at least three set of data to produce p-value. We are very sorry for not being able to provide the quantitation results of all the Western blots.

- As pointed out by the referee, we have added a sentence to explain the asterisk (Figure 1E) in the legend of Figure 1 (line 235).

- In Figure 2C, samples in even lanes are over-loaded based on the amounts of actin levels. In Figure 4B, it was demonstrated that the level of endogenous LAT1 (lanes 1 and 2) was almost the same as the level of HA-tagged LAT1 (lanes 5-10). Therefore, the effects of LAT1 mutants on Akt phosphorylation were not caused by overexpression of LAT1 mutants.

- The main point of Figure 3A is the translocation of LAT1 and recruitment of mTOR to the lysosome. As pointed out by the referee, the levels of LAT1 and mTOR on the lysosome appear to decrease upon PMA treatment. However, we did not consistently observe this effect; instead, LAT1 levels did not change significantly during the early times of PMA treatment. Therefore, we would like to retain the description of the results presented in Figure 3A.

- (Figure 4A): mTORC1 and mTORC2 complexes consists of common subunits and complex-specific subunits, such as Rictor, Raptor and mSIN. It has been recognized that these subunits can also exist in a complex-free form, in addition to their involvement in complex formation. Consequently, it is possible that the levels of subunits within a complex do not exhibit the same behavior. 

- We acknowledge the referee’s point. Unfortunately, however, we accidentally lost the Pan-Akt blot (Figure 4B). We deeply apologize for not being able to fulfill the referee’s request.

- As noted by the referee, we conducted a transwell assay under conditions of Rictor knockdown. These data are presented in Figure 5C and 5D.

Reviewer 2 Report

The manuscript by Tae et al. presents an investigation into the effects of LAT1 expression on mTORC2 activity, offering evidence that LAT1 recruits and activates mTORC2 on the lysosome for PMA-induced cell migration. Overall, the topic is novel and holds significance within the field. The authors' approach is logical and reasonable, as they sequentially demonstrate the localization of LAT1, its interaction with mTORC2 through Rictor, and its impact on PMA-induced cell migration. The paper is well written, with clear presentation of findings. Below please find my specific comments.

Major:

A significant concern pertains to Figure 4A. While Raptor is detected in samples treated with anti-raptor antibodies, it is unclear why mTOR is not detected in the same samples, given that Raptor is a subunit of mTOR. Figure 4A is crucial for establishing the paper's narrative, particularly regarding the interaction between LAT1 and mTORC2 instead of mTORC1. The potential for mTORC1 and mTORC2 to coexist within protein complexes raises questions about whether the co-localization of LAT1 and mTORC2 may be mediated by mTORC1. Clarification on this point is essential to support the argument that LAT1 interacts with mTORC2 and not mTORC1.

Minor:

1. Definition of PMA (line 191) is needed earlier in the text.

2. This manuscript presents evidence of PMA-induced LAT1 translocation to the lysosome and notes that ubiquitination of LAT1 is crucial for its endocytosis. Could this indicate LAT1 degradation via the lysosome or proteasome? Have the authors explored whether the protein expression level of LAT1 changes in response to PMA induction? That may provide deeper insights into the regulatory mechanisms at play.

3. I wonder whether findings on mTORC2-LAT1 interaction are cell-context dependent. Discussing this aspect would enhance the paper's scope.

4. In addition to briefly mentioning LAT1's role in mTORC1 activation, it is suggested that the authors provide a visual representation (e.g., diagram) showcasing LAT1's function in mTORC1 and mTORC2 activation. Such a diagram would emphasize the differences between mTORC1 and mTORC2 and underscore the significance of the findings.

Author Response

RESPONSE TO THE COMMENTS BY REFEREE#2

The manuscript by Tae et al. presents an investigation into the effects of LAT1 expression on mTORC2 activity, offering evidence that LAT1 recruits and activates mTORC2 on the lysosome for PMA-induced cell migration. Overall, the topic is novel and holds significance within the field. The authors' approach is logical and reasonable, as they sequentially demonstrate the localization of LAT1, its interaction with mTORC2 through Rictor, and its impact on PMA-induced cell migration. The paper is well written, with clear presentation of findings. Below please find my specific comments.

Major comment 1: A significant concern pertains to Figure 4A. While Raptor is detected in samples treated with anti-raptor antibodies, it is unclear why mTOR is not detected in the same samples, given that Raptor is a subunit of mTOR. Figure 4A is crucial for establishing the paper's narrative, particularly regarding the interaction between LAT1 and mTORC2 instead of mTORC1. The potential for mTORC1 and mTORC2 to coexist within protein complexes raises questions about whether the co-localization of LAT1 and mTORC2 may be mediated by mTORC1. Clarification on this point is essential to support the argument that LAT1 interacts with mTORC2 and not mTORC1.

Response: The mTORC1 and mTORC2 complexes act on distinct sets of target proteins, which raise the question of how substrate specificity is determined. Current evidence suggests that substrate specificity of mTORC1 and mTORC2 is determined at the level of recruitment. The mTOR kinase itself appears to exercise little discrimination because the truncated mTOR kinase alone (without other subunits) is able to indiscriminately phosphorylate 4E-BP, S6K, Akt in vitro. It has been known that complex-specific subunits, such as Raptor for mTORC1 and mSIN and Rictor for mTORC2, are responsible for binding to substrates (Battaglioni, Benjamin et al. 2022, Yu, Chen et al. 2022). In Figure 4A, relatively fewer amounts of mTOR were detected in the immunoprecipitation using the anti-Raptor antibody than using the anti-Rictor antibody. However, published literature has indicated that the relative amounts of mTORC1 and mTORC2 complexes can vary depending on cell lines. For example, much more mTOR kinase was associated with Rictor in HeLa cells, while more mTOR kinase was associated with Raptor in HEK293 cells. The experiments depicted in Figure 4A were carried out using HeLa cells. Therefore, our results are consistent with published data (Sarbassov, Ali et al. 2004). Hence, we believe that LAT1 associates with mTORC2, not mTORC1, via interaction with Rictor, an mTORC2-specific subunit.

References

Battaglioni, S., D. Benjamin, M. Wälchli, T. Maier and M. N. Hall (2022). "mTOR substrate phosphorylation in growth control." Cell 185(11): 1814-1836.

Sarbassov, D. D., S. M. Ali, D. H. Kim, D. A. Guertin, R. R. Latek, H. Erdjument-Bromage, P. Tempst and D. M. Sabatini (2004). "Rictor, a novel binding partner of mTOR, defines a rapamycin-insensitive and raptor-independent pathway that regulates the cytoskeleton." Curr Biol 14(14): 1296-1302.

Yu, Z., J. Chen, E. Takagi, F. Wang, B. Saha, X. Liu, L. M. Joubert, C. E. Gleason, M. Jin, C. Li, C. Nowotny, D. Agard, Y. Cheng and D. Pearce (2022). "Interactions between mTORC2 core subunits Rictor and mSin1 dictate selective and context-dependent phosphorylation of substrate kinases SGK1 and Akt." J Biol Chem 298(9): 102288.

Minor comment 1: Definition of PMA (line 191) is needed earlier in the text.

Response: As pointed out by the referee, we have added definition of PMA in the introduction section (line 77).

Comment 2. This manuscript presents evidence of PMA-induced LAT1 translocation to the lysosome and notes that ubiquitination of LAT1 is crucial for its endocytosis. Could this indicate LAT1 degradation via the lysosome or proteasome? Have the authors explored whether the protein expression level of LAT1 changes in response to PMA induction? That may provide deeper insights into the regulatory mechanisms at play.

Response: Barthelemy and Andre (2019) reported that LAT1 undergoes lysosomal degradation following endocytosis, with the level of LAT1 gradually decreased over 12 hours.  However, in this study, we optimized the conditions (time and concentration of PMA treatment) for LAT1 localization to the lysosome, and we conducted all experiments 6 hours after PMA treatment. Our study primarily focuses on the effects of lysosomal LAT1 on mTOR-mediated cellular function, rather than solely on the degradation of lysosomal LAT1. In addition, the expressions of endogenous LAT1 and HA-LAT1 were not altered in response to PMA treatment, as shown in Figure 4B (lanes 1 and 2). These same results were also observed in the study by Barthelemy and Andre (2019).

Reference

Barthelemy, C. and B. André (2019). "Ubiquitylation and endocytosis of the human LAT1/SLC7A5 amino acid transporter." Sci Rep 9(1): 16760.

Comment 3. I wonder whether findings on mTORC2-LAT1 interaction are cell-context dependent. Discussing this aspect would enhance the paper's scope.

Response: We observed LAT1-mTORC2 interaction in several different cell lines, including HeLa, HEK293, and A549. Since both LAT1 and mTORC2 are ubiquitously expressed in most cell lines, we believe that LAT1-mTORC2 interaction does not exhibit cell-type specificity. However, it is plausible that LAT1-mTORC2 interaction is induced under specific physiological conditions, such as PKC activation and enrichment of LAT1 localization to the lysosome. Further studies are required to address these questions. 

Comment 4. In addition to briefly mentioning LAT1's role in mTORC1 activation, it is suggested that the authors provide a visual representation (e.g., diagram) showcasing LAT1's function in mTORC1 and mTORC2 activation. Such a diagram would emphasize the differences between mTORC1 and mTORC2 and underscore the significance of the findings.

Response: As pointed out by the referee, LAT1 function in the coordination of mTORC1 and mTORC2 is of great interest. We agree with the referee’s suggestion. Therefore, we have provided a graphical abstract that illustrates the effects LAT1 on mTORC2 for cell migration and on mTORC1 for cell growth and proliferation.

Reviewer 3 Report

In this study Choi and coworkers present results to support LAT1 dependent mTORC2 activation at the lysosome. Authors specifically identify N-terminal lysine residues of LAT1 critical for the endocytic turnover and lysosomal targeting of LAT1. Finally, using a combination of biochemistry and wound healing assay, authors conclude that LAT1 mediated mTORC2 activation allows migration of these cells.

I have several concerns with the study in its current form  and hence can’t recommend an acceptance. I hope authors will find these comments useful to revise their manuscript.

Major concern:

1.     Authors haven’t cited important recent field literature on mTORC2 and its role in cell migration. Please cite these two recent studies in your introduction and discussion - PMID: 36542482 and PMID: 36857159.

2.     This study entirely relies on overexpression of HA-tagged LAT1 and its mutants. Authors should provide an estimate of the overexpression compared to endogenous LAT1 and controls to check if overexpression has any adverse effect of cell proliferation and survival.

3.     Figure 1 images have no scale bar. Please make sure all images have proper scale bar and matched intensity scaling. Please provide zoomed inset of cells to clearly show the colocalization and spatial distribution of the two-color fluorescence in the images (Fig 1 A, C).

4.     Authors should validate their colocalization results obtained with HA-antibody using an antibody against LAT1. In other words, does LAT1 behave the same way upon PMA treatment as these overexpressed constructs.

5.     In PLA assay, why are the signals intracellular for N3KR if the construct is membrane retained ?

6.     Lysosomal activation of mTORC2 is provocative idea but the evidence presented in support of the same are very slim. Can authors validate this result using Locator tool tagged to lysosome ?

7.     Please clarify if Akt S473 phosphorylation is used to monitor mTORC2 activation (Figure 4B). Also show the Pan-Akt blot as a control and provide a quantification of phospho-Akt normalized to Pan-Akt.

8.  Authors should add more evidence using single cell migration assays and/or transwell migration assay to support the wound healing data in Figure 5. Hela and HEK cells don’t really migrate much, and the results presented could be confounded by wound area filled by cell proliferation instead of real migration. In summary, the migration phenotype and claims are exaggerated and showed be removed or supported with other complementary approaches.

Author Response

RESPONSE TO THE COMMENTS BY REFEREE#3

In this study Choi and coworkers present results to support LAT1 dependent mTORC2 activation at the lysosome. Authors specifically identify N-terminal lysine residues of LAT1 critical for the endocytic turnover and lysosomal targeting of LAT1. Finally, using a combination of biochemistry and wound healing assay, authors conclude that LAT1 mediated mTORC2 activation allows migration of these cells.

I have several concerns with the study in its current form and hence can’t recommend an acceptance. I hope authors will find these comments useful to revise their manuscript.

Comment 1. Authors haven’t cited important recent field literature on mTORC2 and its role in cell migration. Please cite these two recent studies in your introduction and discussion - PMID: 36542482 and PMID: 36857159.

Response: As pointed out by the referee, we have included two additional references in the first paragraph of results section (line 343, References #46 and #47).

Comment 2. This study entirely relies on overexpression of HA-tagged LAT1 and its mutants. Authors should provide an estimate of the overexpression compared to endogenous LAT1 and controls to check if overexpression has any adverse effect of cell proliferation and survival.

Response: In Figure 4B, it was demonstrated that the level of endogenous LAT1 (lanes 1 and 2) was nearly identical to the level of HA-tagged LAT1 (lanes 5-10). Therefore, the effects of LAT1 mutants on Akt phosphorylation were not caused by the overexpression of LAT1 mutants.

Comment 3. Figure 1 images have no scale bar. Please make sure all images have proper scale bar and matched intensity scaling. Please provide zoomed inset of cells to clearly show the colocalization and spatial distribution of the two-color fluorescence in the images (Fig 1 A, C).

Response: In response to referee’s suggestion, we have added scale bars in the images of Figure 1, Figure 2 and Figure 3. Furthermore, we have provided images of zoom-in insets in Figure 1 and Figure 2.

Comment 4. Authors should validate their colocalization results obtained with HA-antibody using an antibody against LAT1. In other words, does LAT1 behave the same way upon PMA treatment as these overexpressed constructs.

Response: It was demonstrated in the published data (Barthelemy and André 2019) that endogenous LAT1 and GFP-LAT1 behave the same way upon PMA treatment.

Reference: Barthelemy, C. and B. André (2019). "Ubiquitylation and endocytosis of the human LAT1/SLC7A5 amino acid transporter." Sci Rep 9(1): 16760.

Comment 5. In PLA assay, why are the signals intracellular for N3KR if the construct is membrane retained?

Response: As observed in most biological phenomenon, biological alterations are not all-or-non. Therefore, it is likely that a small portion of the N3KR mutant could translocate to the lysosome, where N3KR colocalizes with the mTORC2 complex to a lesser extent compared to the colocalization of WT LAT1 with mTORC2.

Comment 6. Lysosomal activation of mTORC2 is provocative idea but the evidence presented in support of the same are very slim. Can authors validate this result using Locator tool tagged to lysosome?

Response: Several different tools are utilized to study the lysosomal function of proteins. Lysosomal immunoprecipitation (Lyso-IP) is one of the methods to investigate lysosomal localization and protein function, and it has been employed in numerous studies published in top journals. I agree with the referee’s viewpoint that relying on a single method may not be sufficient to draw a conclusive result. However, each method comes with its own merits and disadvantages. Therefore, we employed both a biochemical approach, such as Lyso-IP, and a molecular imaging approach, such as the Proximity Ligation Assay (PLA). We believe that the Locator tool approach is in line with Lyso-IP.

Comment 7. Please clarify if Akt S473 phosphorylation is used to monitor mTORC2 activation (Figure 4B). Also show the Pan-Akt blot as a control and provide a quantification of phospho-Akt normalized to Pan-Akt.

Response: We acknowledge the referee’s point. Unfortunately, however, we accidentally lost the Pan-Akt blot. We deeply apologize for not being able to fulfill the referee’s request.

Comment 8. Authors should add more evidence using single cell migration assays and/or transwell migration assay to support the wound healing data in Figure 5. Hela and HEK cells don’t really migrate much, and the results presented could be confounded by wound area filled by cell proliferation instead of real migration. In summary, the migration phenotype and claims are exaggerated and showed be removed or supported with other complementary approaches.

Response: As suggested by the referee, we have added the results of the transwell migration assay using A549 cells (Figure 5C and 5D).

Round 2

Reviewer 1 Report

The authors have addressed some points raised by this Reviewer. However, other points remain to be clarified:

The authors have quantified western blots shown in Figure 1E (Point 1) and 2C (Point 2). However, quantifications for western blots shown in Figure 2D (Point 2), 3A (Point 3), 4A-B (Point 4) have not been provided. The authors state that results were reproducible over 2-3 experiments, but statistical analysis was not possible as the same set of experiment (lane by lane) was not available and therefore cannot be analyzed statistically (Fig 2D, 3A, 4A) and/or the blot was lost (Figure 4B). If certain experiments were performed 3 times, and appropriate controls were included, quantification and some statistical analysis of key samples should be feasible. Alternatively, a repeat should also be considered. Likewise, if a blot was lost, stripping and reprobing of other filters (e.g. the P-Akt membrane) and/or repeating the experiment should be considered.

Importantly, these missing quantifications cover major findings of this manuscript:

Fig. 2D: Plasma membrane association of WT-LAT1 and C4KR-LAT1, but not N3KR-LAT1, is reduced after PMA treatment. This critical experiment to highlight that the N-terminus of LAT1 is ‘crucial’ as stated by the authors and needs quantification.

Fig. 3A: Recruitment of mTOR to lysosomes upon PMA treatment. According to the author's response, if LAT1 and mTOR did not consistently show reduced association with lysosomes at 3-6hrs, this should become evident with a quantification of the multiple experiments that appear available to the authors.

Fig. 4A: LAT1 association with Rictor increased with PMA. This important findings needs quantification to support its relevance.

Fig. 4B: LAT1 knockdown efficiency appears different between samples, in some samples probably only approx. 50%? When comparing HA-LAT1 and LAT1 signal intensities, how do the authors know that the LAT1 antibody recognizes endogenous and ectopically expressed HA-LAT1 with the same affinity? Without providing total Akt levels is, drawing conclusions from P-Akt levels could be misleading. Evidence that these experiments are reproducible should be provided.

As part of the Revision (Point 5), the authors now provide a new panel in Fig. 5C showing a transwell assay with cells transfected with siLAT1 and siRictor. Knockdown efficacy of siLAT1 and in particular siRictor by western blot needs to be provided here.

Author Response

The authors have addressed some points raised by this Reviewer. However, other points remain to be clarified:

The authors have quantified western blots shown in Figure 1E (Point 1) and 2C (Point 2). However, quantifications for western blots shown in Figure 2D (Point 2), 3A (Point 3), 4A-B (Point 4) have not been provided. The authors state that results were reproducible over 2-3 experiments, but statistical analysis was not possible as the same set of experiment (lane by lane) was not available and therefore cannot be analyzed statistically (Fig 2D, 3A, 4A) and/or the blot was lost (Figure 4B). If certain experiments were performed 3 times, and appropriate controls were included, quantification and some statistical analysis of key samples should be feasible. Alternatively, a repeat should also be considered. Likewise, if a blot was lost, stripping and reprobing of other filters (e.g. the P-Akt membrane) and/or repeating the experiment should be considered. Importantly, these missing quantifications cover major findings of this manuscript:

Fig. 2D: Plasma membrane association of WT-LAT1 and C4KR-LAT1, but not N3KR-LAT1, is reduced after PMA treatment. This critical experiment to highlight that the N-terminus of LAT1 is ‘crucial’ as stated by the authors and needs quantification.

Fig. 3A: Recruitment of mTOR to lysosomes upon PMA treatment. According to the author's response, if LAT1 and mTOR did not consistently show reduced association with lysosomes at 3-6hrs, this should become evident with a quantification of the multiple experiments that appear available to the authors.

Fig. 4A: LAT1 association with Rictor increased with PMA. This important findings needs quantification to support its relevance.

Fig. 4B: LAT1 knockdown efficiency appears different between samples, in some samples probably only approx. 50%? When comparing HA-LAT1 and LAT1 signal intensities, how do the authors know that the LAT1 antibody recognizes endogenous and ectopically expressed HA-LAT1 with the same affinity? Without providing total Akt levels is, drawing conclusions from P-Akt levels could be misleading. Evidence that these experiments are reproducible should be provided.

As part of the Revision (Point 5), the authors now provide a new panel in Fig. 5C showing a transwell assay with cells transfected with siLAT1 and siRictor. Knockdown efficacy of siLAT1 and in particular siRictor by western blot needs to be provided here.

Response:

- As pointed out by the referee, we have added pan-Akt blot image in Figure 4B and provided densitometric quantitation results for Figure 2D, 3A, 4A, and 4B. In addition, Western blot images have been included in Figure 5D to display the knockdown efficiency of LAT1 and Rictor.

- We believe that these additions fulfill all the requirements requested by the reviewer.

Reviewer 3 Report

Authors have not able to address all the concerns of this reviewer! Especially for comment 7; authors acknowledge they have lost the original data for the critical control sought by this reviewer. 

Authors don't provide convincing arguments to address comment 3 and make hand-waving half-baked suggestions!

Author Response

Authors have not able to address all the concerns of this reviewer! Especially for comment 7; authors acknowledge they have lost the original data for the critical control sought by this reviewer. 

Response: As pointed out by the referee, we have included pan-Akt blot and quantitated the Western blot results.

Authors don't provide convincing arguments to address comment 3 and make hand-waving half-baked suggestions!

Response: In response to referee’s suggestion for comment 3, we have added scale bars in the images of Figure 1, Figure 2 and Figure 3. Furthermore, we have provided images of zoom-in insets in Figure 1 and Figure 2. We believe that we have made all the necessary corrections as per the referee’s request.

Round 3

Reviewer 1 Report

The authors have addressed the points raised by this reviewer.

Only mInor editing might be required.

Reviewer 3 Report

Authors have made adequate revision to the manuscript!